# An origami paper-based nanoformulated immunosensor detects picograms of VEGF-C per milliliter of blood

Shuai Sun[1,2,4], Yang Wang[1,2,4], Tao Ming[1,2], Jinping Luo[1,2], Yu Xing[1,2], Juntao Liu[1,2], Ying Xiong[3], Yuanyuan Ma[3], Shi Yan[3], Yue Yang[3] & Xinxia Cai[1,2 ✉]

Detecting vascular endothelial growth factor C (VEGF-C), a kind of tumor biomarker, is of significant clinical importance in evaluating the prognosis of patients with cancer. However, laboratory analyses are usually not suitable for point-of-care testing because they are expensive and time consuming. In response to these challenges, we fabricated an origami paper-based microfluidic electrochemical device. To improve the specificity of VEGF-C detection, nanocomposites, synthesized by new methylene blue (NMB), amino-functional single-walled carbon nanotubes (NH$_2$-SWCNTs), and gold nanoparticles (AuNPs), were used to modify the surface of working electrodes. Results of electrochemical detection showed that the immunosensor had excellent linearity, ranging from 0.01 to 100 ng mL$^{-1}$ ($R^2 =$ 0.988), and the limit of detection was 10 pg mL$^{-1}$. To confirm the high specificity of the device under real-world conditions, we evaluated the device using clinical serum samples from our hospital. The results demonstrated that the device had an excellent performance and could provide a platform for real-time detection of cancers.

[1] State Key Laboratory of Transducer Technology, Institute of Electronics, Chinese Academy of Sciences, Beijing 100190, China. [2] University of Chinese Academy of Sciences, Beijing 10090, China. [3] Key Laboratory of Carcinogenesis and Translational Research (Ministry of Education), Department of Thoracic Surgery II, Peking University Cancer Hospital & Institute, Beijing 100142, China. [4]These authors contributed equally: Shuai Sun, Yang Wang.
✉email: xxcai@mail.ie.ac.cn

Vascular endothelial growth factor C (VEGF-C), which belongs to the VEGF family, is a major inducer of angiogenesis under both normal and pathological conditions[1–3]. With a high degree of homology to VEGF, it plays an important role in regulating the formation of the lymphatic vessels[4]. Serum concentrations of VEGF-C are typically in the nanogram per milliliter range[5]. However, overexpression of VEGF-C is observed in various malignant tumors[6,7]. Thus, determining VEGF-C levels could be useful for predicting advanced disease and individuals at risk for cancer[8,9].

To date, only a few studies on the detection of VEGF-C have been reported, and immunohistochemistry is the most widely accepted method of detecting VEGF-C in clinicopathological circumstances[10,11]. Because determining VEGF-C by immunohistochemistry is time consuming, is complex, and requires complicated preprocessing of blood samples, alternative technologies have been developed based on interactions between antibodies and antigens[12]. For example, Duff et al. proposed using quantitative indirect enzyme-linked immunosorbent assay (ELISA) with goat polyclonal anti-human VEGF-C antibody to detect VEGF-C both in normal individuals and in patients with colorectal cancer in plasma. The assay was highly sensitive and enabled detection at a range of 0.4–100 U mL$^{-1}$ with less than 8% variation[1]. Cai et al. reported a resistive-pulse biosensor using antibody-modified silver nanoparticles (AuNPs) and nanopipettes to detect VEGF-C; its main challenge was to differentiate AuNPs with attached antibodies for VEGF-C and antigen-conjugated particles, as they were of similar size[13]. Although these methods are relatively simple and easy to perform for point of care testing (POCT), they all showed an inefficient performance. Therefore, the development of a cost-effective and rapid means of detecting VEGF-C is urgently needed.

Microfluidic paper-based analytical devices (μPADs), which are fabricated from paper, have attracted lots of research attention since they were first introduced by Whitesides and colleagues in 2007[14,15]. As a substrate material, paper has many unique advantages over traditional materials, including low cost[16], good biocompatibility[17], and power-free fluid transport via capillary action[18]. What is more, it is easy to fabricate such a device by either creating barriers within the paper itself or selectively cutting and removing paper without the need for an ultraclean room[19,20]. As a result, μPADs are good alternatives when low cost and portability are critical[21,22]. Researchers have used different detection methods to conduct chemical measurements based on paper microfluidics[23–25]. Colorimetry is the method most widely used with μPADs, and results of these assays are usually quantified by comparing the intensity of the color with the naked eye or by camera, which lacks the sensitivity and selectivity of traditional analytical instrumentation[26,27]. Thus, a detection method with high specificity and sensitivity is needed to determine low levels of analytes in complex biological samples, including plasma and urine. Electrochemical detection is an attractive alternative method because of its high sensitivity, high selectivity, and quick response with the proper choice of detection potential and electrode material[28,29]. An additional advantage of electrochemical detection is its simplicity of instrumentation, which requires little electrical power in the field[30]. Thus, many researchers used electrochemical methods to detect biochemical substances and made an excellent progress, such as hydrogen peroxide, glucose, and biomarkers[31–33]. In our previous work, we fabricated a microfluidic paper-based analytical device for the highly sensitive detection of carcinoembryonic antigen (CEA). The device enabled electrochemical detection using graphene-based nanocomposites modified screen-printed carbon electrodes[34].

Carbon nanotubes were first discovered in 1991 and have received much attention in the ensuing years[35]. Depending on the conditions in which they are formed, they can be categorized as either multi-walled coaxial tubules (multi-walled carbon nanotubes; MWCNTs) or bundles consisting of individual cylinders (single-walled carbon nanotubes; SWCNTs)[36,37]. Experimental results have shown that CNTs possess extremely high electron transfer capacity as well as a high Young's modulus[38]. Given their excellent properties, they have been widely used in the modification of various biosensors[39,40]. What is more, the introduction of nanomaterials has made a great contribution to the boost development of μPADs with excellent analytical properties[41].

Here we present electrochemical detection for a microfluidic paper-based analytical device for highly sensitive point-of-care detection of VEGF-C in serum samples. Our previous working electrode was coated with as-synthesized new methylene blue (NMB)/NH$_2$-SWCNTs/AuNPs nanocomposites to introduce electrochemical signals and improve electrode conductivity. Working electrodes were subsequently coated with VEGF-C antibodies for the detection of VEGF-C in human serum samples. A real-time electrochemical detection method, which avoided labeling biometric molecule, was used to realize the goal of POCT. The premise behind the detection method was that the formation of an insulating antigen and antibody compound on the electrode would hinder the electrons transfer efficiency, which in turn would result in a lower current. Results of electrochemical detection showed that the immunosensor had excellent linearity, ranging from 0.01 to 100 ng mL$^{-1}$ ($R^2 = 0.988$), and the limit of detection was 10 pg mL$^{-1}$. What is more, the immunosensor showed a good performance in repeatability, selectivity, and specificity. The fabricated electrochemical paper-based immunosensor, with its high sensitivity and low cost, provides an alternative platform for early cancer detection, in particular in less developed areas.

## Results

**Characterization of synthesized NMB/NH$_2$-SWCNT/AuNPs.** We used transmission electron microscopy (TEM) and scanning electron microscopy (SEM) to observe properties of the NMB/NH$_2$-SWCNT/AuNP nanocomposites. A TEM image is shown in Fig. 1a. SWCNT, which had a tubular structure, is shown. SWCNT nanomaterial increased the current response by reducing the electrode impedance. Because of the interactions between benzene rings, NMB molecules could combine with SWCNT by π-π stacking interactions. Meanwhile, circular particles adhered evenly to the SWCNT. This suggests that AuNPs were uniformly distributed on single-walled carbon nanotubes without aggregation by the connection between Au and amino groups. The AuNPs provided sites for antibodies to attach. This indicates that synthesis could lead to excellent performance of nanocomposites. Given the great combination of nanocomposites, the device had good electrochemical properties.

SEM images are shown in Fig. 1b and Fig. 1c. As shown in Fig. 1b, after modification of the nanocomposites, the surface of the working electrode had a rough structure. The increase in the specific surface area led to a decrease in current impedance, which was advantageous for signal transmission. Figure 1c shows an enlarged SEM image, which shows the nanoporous structure more clearly.

**Electrochemical properties of the VEGF-C immunosensor.** The CV and DPV responses were measured with an autolab instrument in 0.1 M PBS solution (pH = 7.4). The CV responses are shown in Fig. 2a. From this graph, we can see that the bare electrode had no response (curve a5). This means that bare electrodes had no oxidation or reduction without nanocomposite modification. By contrast, with NMB/NH$_2$-SWCNT/AuNP nanocomposite modification,

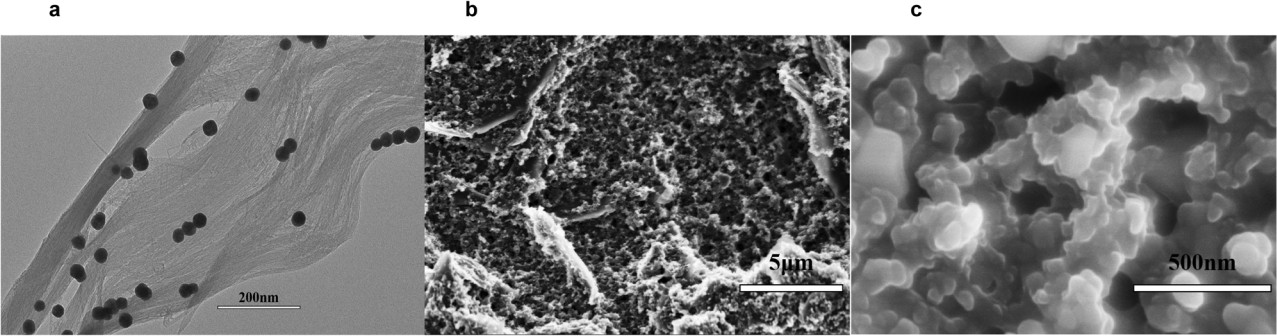

**Fig. 1 Characterization of synthesized NMB/NH₂-SWCNT/AuNPs. a** Transmission electron microscopy (TEM); **b**, **c** Scanning electron microscopy (SEM) images of the NH₂-SWCNT/NMB/AuNP nanocomposites.

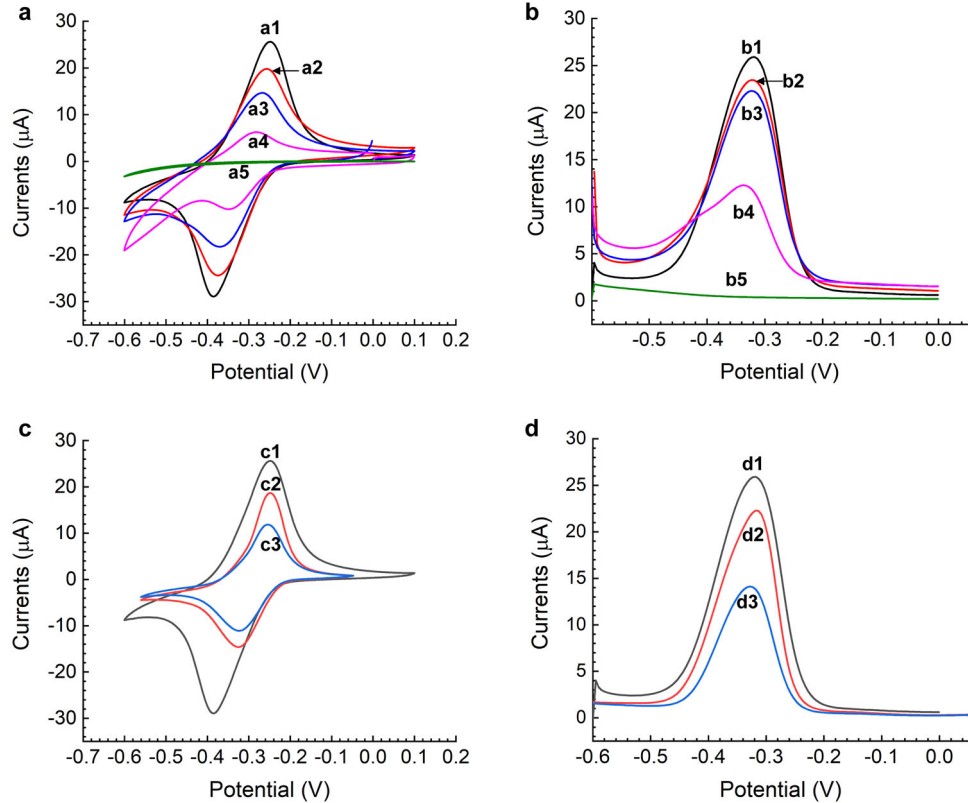

**Fig. 2 Electrochemical properties of the VEGF-C immunosensor and the nanocomposites assembly process. a** CV responses of the device: (a1) CV responses of NH₂-SWCNT/NMB/AuNP nanocomposites modified working electrode; (a2) The working electrode coated with VEGF-C antibody; (a3) The working electrode immobilized with BSA blocking solution; (a4) The immunosensor incubated with 500 pg mL$^{-1}$ VEGF-C antigen; (a5) The bare working electrode. **b** DPV responses of the device: (b1) The working electrode modified with NH₂-SWCNT/NMB/AuNP nanocomposites; (b2) The working electrode coated with VEGF-C antibody; (b3) The working electrode immobilized with BSA blocking solution; (b4) The immunosensor incubated with 500 pg mL$^{-1}$ VEGF-C antigen; (b5) The bare working electrode. **c** CV responses of the device: (c1) CV responses of NH₂-SWCNT/NMB/AuNP nanocomposites modified working electrode; (c2) The working electrode immobilized with NH₂-SWCNT/NMB composites; (c3) The working electrode coated with NMB only. **d** DPV response of the device: (d1) The working electrode modified with NH₂-SWCNT/NMB/AuNP nanocomposites; (d2) The working electrode immobilized with NH₂-SWCNT/NMB composites; (d3) The working electrode coated with NMB only.

the electrodes showed better electrochemical performance (curve a1), which could suggest that electrode modification was effective. Affected by NMB, the oxidation peak and reduction peak were obtained at −300 and −360 mV, respectively. Because VEGF-C antibodies affected the electron transfer of electroactive substances on the surface of the electrodes, the peak current decreased steadily with VEGF-C antibodies (curve a2). Similarly, when BSA blocking solution was added to the surface of the electrodes, the responses also decreased gradually (curve a3). Subsequently, the peak current

fell in pace with the continuous addition of protein (curve a4), which confirms that the specific binding of antigen and antibody also had an effect on the electron transfer of electroactive substances. The higher the concentration of protein, the greater the effect on the signal. The CV results suggest great performance of the immunosensor.

The obtained DPV responses are shown in Fig. 2b. The responses were measured under the same conditions as the CV responses. As shown in Fig. 2b, when NMB/NH₂-SWCNT/AuNP

composites were modified, the working electrodes (curve b1), whose peak current was 25.91 μA, performed better than the bare electrodes (curve b5). Then, when we added the anti-VEGF-C antibody, the peak current fell to 23.46 μA (curve b2). Next, after we dropped the BSA blocking solution, the electrodes' responses steadily decreased to 22.32 μA (curve b3). Because of the influence of the VEGF-C antigen, with the dripping of 500 pg mL$^{-1}$ VEGF-C, the electrodes showed a dramatic decrease in the peak current (curve b4). The CV curve and DPV curve fully verify the correctness of our experimental premise. Accordingly, they confirm that modification of the paper-based immunosensor was successful for testing.

To characterize the assembly process of VEGF-C sensor, we measured the CV and DPV responses of the immunosensors which were modified with NMB, NMB/NH$_2$-SWCNT, and NMB/NH$_2$-SWCNT/AuNP, respectively. The CV responses are shown in Fig. 2c. From this graph, we can see that the working electrode had an electrochemical signal due to the fact of NMB (curve c3). By contrast, after adding the NH$_2$-SWCNT nanomaterial, the electrode showed a better electrochemical performance, which could suggest that NH$_2$-SWCNT could reduce the resistance of the sensor (curve c2). Similarly, when AuNP was assembled on the nanocomposites, the peak current also increased gradually (curve c1).

The obtained DPV responses are shown in Fig. 2d. The responses were measured under the same conditions as the CV responses. As shown in Fig. 2d, when the NMB was modified on the working electrode, the sensor had a redox signal (curve d3). When NH$_2$-SWCNT was connected to NMB, the peak current rose to 22.3 μA (curve d2). Then, when AuNP was assembled on the nanocomposites, the working electrode (curve d1), whose peak current was 25.91 μA, performed better than the sensor modified by NH$_2$-SWCNT/NMB composites, which confirmed that the gold nanoparticles could improve the detection sensitivity of the sensor.

**Optimization of experimental conditions for the immunosensor.** During the experiment, we found that the concentration of NMB, the ratio of NMB/SWCNT to AuNPs, and the concentration of anti-VEGF-C antibody affected the response of the paper-based device. To determine the best concentration of NMB, we prepared different concentrations of NMB solution, ranging from 0.5 to 8 mg mL$^{-1}$. We mixed the different concentrations of NMB and SWCNT at a ratio of 1:1. After that, we determined their responses, which are shown in Fig. 3a. From this graph, we can see that the device performed best when the concentration of NMB was 2 mg mL$^{-1}$.

The amount of AuNPs also played an important role in the performance of the immunosensor. As shown in Fig. 3b, AuNPs were mixed with NMB/SWCNT nanocomposites at ratios of 1:1, 2:1, 5:1, 8:1, and 10:1. We observed that when the ratio increased to 5:1, the peak current reached its maximum, which confirms that 5:1 was the best proportion of AuNPs and NMB/SWCNT.

To optimize the concentration of the antibody, we prepared the antibody mother liquor in a concentration of 1 μg mL$^{-1}$. After that, we diluted the antibody to different concentrations, ranging from 0.031 to 1 μg mL$^{-1}$. Then we added antibodies on the surface of several electrodes that were modified by NMB/NH$_2$-SWCNT/AuNP nanocomposites. After CV and DPV scanning in the PBS solution (pH = 7.4), the results were recorded and are shown in Fig. 3c. The results suggested that as the dilution multiple of the anti-VEGF-C antibody increased, the DPV response of the device increased. When the antibody was diluted to 8 times (0.125 μg mL$^{-1}$), the peak current of the DPV curve achieved its maximum response. Based on this, we eventually selected 0.125 μg mL$^{-1}$ anti-VEGF-C antibody for subsequent experiments.

**Repeatability and selectivity of the VEGF-C immunosensor.** The selectivity of an immunosensor can reflect the anti-interference ability of the device, which is a critical factor for assessing the performance of the device. To test the selectivity of the immunosensor, we used several interfering substances, including epidermal growth factor receptor (EGFR), programmed death ligand 1 (PD-L1), carcinoembryonic antigen (CEA), neuron-specific enolase (NSE), ascorbic acid (AA), and uric acid (UA). To begin the experiment, we mixed 1 ng mL$^{-1}$ VEGF-C protein with 10 ng mL$^{-1}$ of the interfering substance. After that, we determined the DPV response with the VEGF-C immunosensor. The results are shown in Fig. 4a. Current variation in the interfering substances influence was 7.88%, −4.55%, 2.48%, 3.04%, −3.18%, and 1.16%, respectively. Because EGFR is a membrane protein present in a low concentration in the serum, the impact of EGFR was not great. The impact of the remaining interferers was within 4.55%, which means that the immunosensor showed great selectivity.

The repeatability of the immunosensor played an important role in our evaluations of the device. We characterized the repeatability of the immunosensor by determining 500 pg mL$^{-1}$ VEGF-C protein with three different devices. The results are shown in Fig. 4b. The coefficient of variation (CV) of the device was 0.67%.

**Analytical performance of the VEGF-C immunosensor.** The electrochemical response of the immunosensor indicated its

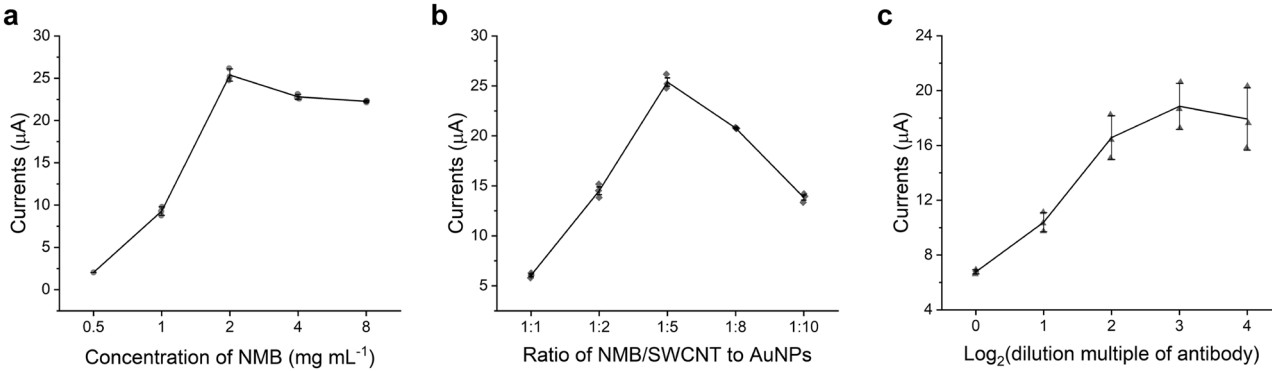

**Fig. 3 Optimization process of the VEGF-C immunosensor. a** Effects of different concentrations of new methylene blue, the best concentration of NMB was 2 mg mL$^{-1}$. **b** Effects of the proportion of NMB/SWCNT to AuNPs, the best ratio of NMB/SWCNT to AuNPs was 1:5. **c** Influence of the dilution multiple of the VEGF-C antibody, the best concentration of the VEGF-C antibody was 0.125 μg mL$^{-1}$.

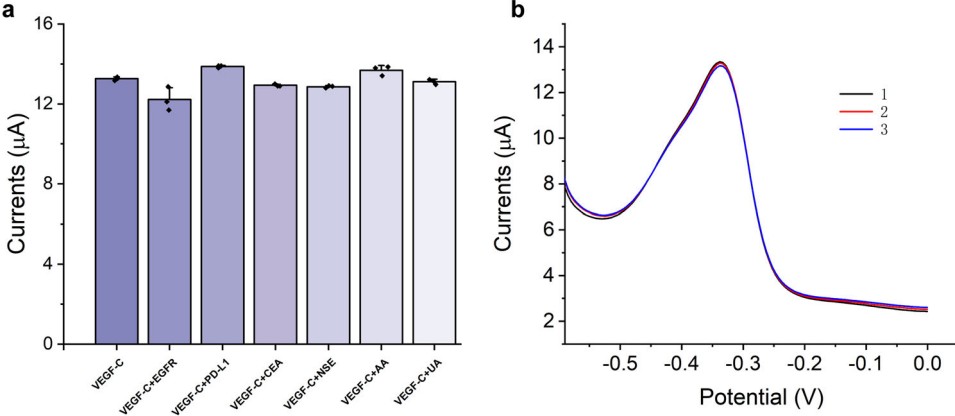

**Fig. 4 Selectivity and repeatability performance of the immunosensor. a** DPV responses of the paper-based immunosensor to 500 pg mL$^{-1}$ VEGF-C and 1 ng mL$^{-1}$ VEGF-C mixed with 10 ng mL$^{-1}$ interfering substances, including EGFR, PD-L1, CEA, NSE, AA, and UA, at a ratio of 1:1. **b** Repeatability of the immunosensor to 500 pg mL$^{-1}$ VEGF-C.

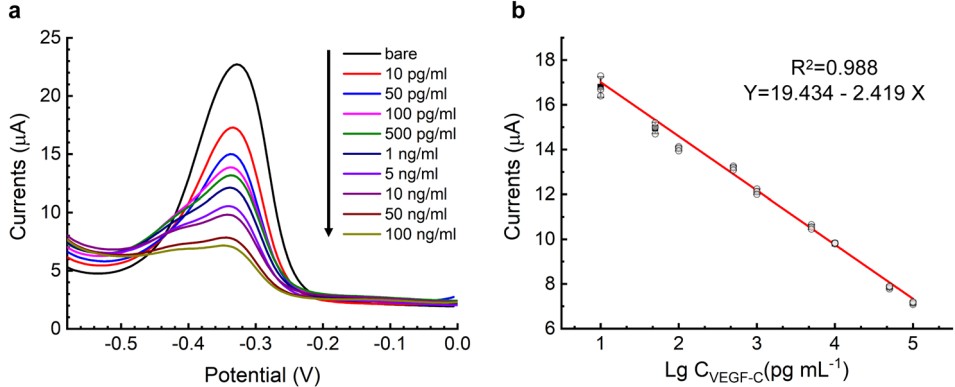

**Fig. 5 Assay results of standard VEGF-C solutions in 0.1 M PBS solution (pH = 7.4). a** DPV responses of the paper-based device to different concentrations of VEGF-C antigen, ranging from 10 pg mL$^{-1}$ to 100 ng mL$^{-1}$. **b** Calibration curve of the peak current of the DPV response and the logarithm concentration of VEGF-C.

performance for detecting the concentration of the VEGF-C protein. As shown in Fig. 5a, DPV responses decreased gradually when the concentration of VEGF-C protein increased. Electron transfer on the surface of the electrodes was affected by the antigen–antibody immunocomplex, which led to a decrease in DPV responses. The x-axis of Fig. 5b shows the logarithm concentration of the VEGF-C antigen, and the y-axis displays the DPV peak current. The calibration plot shows an excellent linear relationship, whose $R^2$ was 0.988 at the concentration 10 pg mL$^{-1}$ to 100 ng mL$^{-1}$. The regression equation of the calibration plot was $I_{dpv}$ (uA) = 19.434 – 2.419 lg $C_{VEGF-C}$ (pg mL$^{-1}$). The limit of detection (LOD) of the immunosensor, which was obtained by actual measurement, was 10 pg mL$^{-1}$.

**Stability of the VEGF-C immunosensor**. One of the advantages of the paper-based device is its low cost. Thus, the stability of the VEGF-C immunosensor is particularly important. To determine the stability of the device, we produced several sensors. After preliminary preparation, the devices were stored at 4 °C. Then we chose one paper-based device each week and measured its DPV response with 1 ng mL$^{-1}$ VEGF-C antigen. Until the peak current dropped 20%, we considered the device valid. The results are shown in Fig. 6. In the first 4 weeks, the peak current remained steady. Subsequently, the response decreased gradually. In the sixth week, the peak current fell to 85.23% of the initial signal. After storage for 7 weeks, the response fell to 70.30%, which

meant that the device was invalid. Therefore, we believe that the device has a lifetime of 6 weeks.

**Analytical results of the VEGF-C clinical serum samples**. To prove that the instrument has practical applications, we tested clinical serum samples with the VEGF-C immunosensor and compared the results to those detected by a commercial instrument from the hospital. All clinical serum samples were provided by Peking University Cancer Hospital & Institute with VEGF-C concentrations ranging from 4.06 to 14.66 ng mL$^{-1}$. The comparison of results is shown in Table 1. The relative deviation between the two methods was less than 9.81%, which confirms that the immunosensor was able to achieve clinical detection. The results also demonstrate that the paper-based device had an excellent specificity of detection. This method represents a platform for the real-time detection of cancers.

**Comparison of performance of sensors for the VEGF detection**. According to Table 2, we compared the performance of our immunosensor to other VEGF detection devices. The results showed that we had made some progress in linear range, LOD, and relative deviation. It may demonstrate that the origami paper-based immunosensor proposed in this work enabled a relative wide linear range, with similar limit of detection.

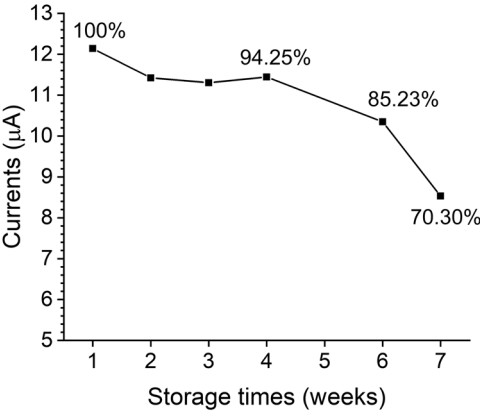

**Fig. 6 The performance of stability of the paper-based immunosensor.** The figure showed the lifetime of several devices stored at 4 °C before the devices were invalid. When the peak current dropped 20%, we considered the device valid.

**Table 1 Assay results of clinical serum samples using the proposed VEGF-C immunosensor and a commercial instrument.**

| No. | Currents (μA) | Proposed immunosensor (ng/mL) | Reference concentration (ng/mL) | Relative deviation (%) |
|-----|---------------|-------------------------------|---------------------------------|------------------------|
| 1 | 8.91 | 5.77 | 5.31 | 8.72 |
| 2 | 8.72 | 6.65 | 6.06 | 9.81 |
| 3 | 8.57 | 7.30 | 7.86 | −7.09 |
| 4 | 8.51 | 7.59 | 8.31 | −8.72 |
| 5 | 8.07 | 9.56 | 8.81 | 8.56 |
| 6 | 8.34 | 8.36 | 9.01 | −7.22 |
| 7 | 8.27 | 8.66 | 9.56 | −9.44 |
| 8 | 7.86 | 10.53 | 9.91 | 6.27 |
| 9 | 8.09 | 9.46 | 10.01 | −5.48 |
| 10 | 7.69 | 11.29 | 10.36 | 9.01 |
| 11 | 7.56 | 11.84 | 11.21 | 5.63 |
| 12 | 7.41 | 12.54 | 12.06 | 3.97 |
| 13 | 7.44 | 12.39 | 13.31 | −6.93 |

## Discussion

**Table 2 Comparison results of the proposed VEGF-C immunosensor and other sensors for detection of VEGF.**

| No. | Substance | Linear range (pg mL$^{-1}$) | LOD (pg mL$^{-1}$) | Relative deviation (%) | Reference |
|-----|-----------|------------------------------|---------------------|------------------------|-----------|
| 1 | VEGF | 10–70 | 10 | 5.0 | 42 |
| 2 | VEGF | 0–400 | 100 | 10 | 43 |
| 3 | VEGF | 5–125 | 5 | 1.47 | 44 |
| 4 | VEGF$_{165}$ | 10–800 | 12 | | 45 |
| 5 | VEGF-C | 0.4–100U | 0.4U | 8.0 | 13 |
| 6 | VEGF-C | 10–100,000 | 10 | 0.67 | This work |

In this work, we fabricated a low-cost electrochemical immunosensor based on an origami paper-based device for highly sensitive detection of VEGF-C. We selected three kinds of materials to synthesize the bio-sensitive membrane, including new methylene blue (NMB), single-walled carbon nanotubes (SWCNT), and gold nanoparticles (AuNPs). NMB can introduce electrochemical currents for the immunosensor due to its redox characteristics. SWCNT is used because of the following two reasons: (i) it could improve the detection sensitivity due to its large specific surface area and good electrical conductivity; (ii) it could help immobilize NMB through π-π stacking interactions. In addition, the AuNPs are introduced not only for its fast electron transportation but also for the interactions between amino groups of VEGF-C antibodies and gold. In the end, these three materials are synthesized together to form a functionalized nanomaterial. After antibody modification, the immunosensor has the ability to detect VEGF-C.

We used the prepared immunosensor for basic performance testing. The results demonstrated that the immunosensor enabled a linearity range from 10 pg mL$^{-1}$ to 100 ng mL$^{-1}$ for VEGF-C standard solutions, with a LOD of 10 pg mL$^{-1}$ at a signal-to-noise ratio of 3. The electrochemical immunosensor also exhibited good selectivity, repeatability, and stability. To make the sensor have a relative good performance, we optimized the experimental conditions for the immunosensor. The results showed that the immunosensor had a good response when the concentrations of NMB and the anti-VEGF-C antibody were 2 mg mL$^{-1}$ and 0.125 μg mL$^{-1}$, respectively. And the proportion of AuNPs and NMB/SWCNT was 5:1. What is more, we further validated the reliability of the immunosensor by assaying clinical serum samples with certain concentrations of VEGF-C. compared with other VEGF protein family biosensors, the paper-based immunosensor had a better performance. In conclusion, compared to other platforms, the proposed immunosensor has the following three advantages. First, origami was used for sample preparation and electrochemical detection, which makes the device much easier to use. In addition, the design further reduces production costs by decreasing the amount of printing masks during screen printing, as all electrodes are printed on one side. Second, NMB/NH$_2$-SWCNT/AuNP nanocomposites were newly synthesized to modify the paper-based device. Experimental results showed that the device exhibited excellent electrochemical properties after modification and could be used for further detection. Third, a label-free detecting device was used that avoided sample pre-treatment, which makes the device suitable for point-of-care testing.

In the future, we will focus on the following three research directions. Firstly, due to the fact that multi parameter detection can improve the accuracy of cancer diagnosis, it is more meaningful to detect multi tumor biomarkers simultaneously to realize the clinical diagnosis function. According to the detection principle we used in this work, our device could be suitable for detection various biomarkers by replacing the bio-recognition elements. As a result, if we combine the multi-parameter paper-based device with different modification processes, multi-parameter detection could be realized.

Secondly, antibodies are used as bio-recognition elements in this work, however, antibodies are usually expensive and thus lead to the high cost of these sensors in usage and maintenance. It will be helpful to reduce the fabrication costs by using aptamers as the recognition elements instead of antibodies, as they are much cheaper, stable and easier to modify.

Finally, the goal of point-of-care testing would be fully realized if we combine the low-cost disposable paper-based device with commercially available portable electrochemical detectors. What's more, the proposed device will be more suitable for early cancer detection at home or in resource-limited areas. However, the performance of the immunosensor is only verified by clinical serum samples in this work, as for the on-site quick testing, the device will need to integrate the serum separation function for the whole blood detection.

## Methods

**Apparatus**. An Autolab electrochemical workstation (PGSTAT302N, Herisau, Switzerland) was used to perform electrochemical measurements, including differential pulse voltammetry (DPV) and cyclic voltammetry (CV). Ultrapure water was processed by an ultrapure water instrument (Michem, China). To create the hydrophobic region, we used a Xerox Color Qube 8570 digital wax printer. Other apparatuses, such as an ultrasonic generator (KH2200E; Hechuang, China), a transmission electron microscope (HT7700; Hitachi, Japan), and an oven (DZF-6020MBE; Boxun, Shanghai), were also used.

**Materials and reagents**. NH$_2$-SWCNT was provided by Nanjing Xianfeng Company (XF-NANO, China). New methylene blue (NMB) was bought from Bailingwei Company (Beijing, China). Anti-VEGF-C antibody came from Beijing Biolink Biotechnology. Human VEGF-C protein was purchased from Sino Biological (Beijing, China). Phosphate-buffered saline (PBS) was prepared with a PBS tablet (Sigma, USA). Bovine serum albumin (BSA) was purchased from Beijing Chemical Reagents Company, China. Chromatography Paper (Whatman No.1) was purchased from Shanghai Wishes Biotechnology (Shanghai, China). Other materials included conductive carbon ink (Acheson-ED581ss, USA) and conductive Ag|AgCl ink (Yingman, CNC-01, China). All chemicals were of

analytical reagent grade and were used without further purification. Finally, Peking University Cancer Hospital & Institute provided us clinical serum samples with certain concentration of VEGF-C.

**Fabrication of the origami paper-based device**. A basic illustration of the origami device is displayed in Fig. 7a. The substrate of the paper-based biosensor was made of Whatman No. 1 Chromatography Paper. To fabricate the microfluidic channels, we used wax printing technology to divide the filter paper into hydrophobic (color) and hydrophilic (white) regions. All hydrophobic illustrations were patterned on a Xerox digital wax printer. After wax printing, the filter paper was placed in an oven (120 °C) for 3 min so the wax liquor would soak the paper. Subsequently, we used screen-printing technology to fabricate the electrodes. The working electrodes (WE 1 and WE 2) and counter electrodes (CE 1 and CE 2) were made of carbon ink, whereas the reference electrodes (RE 1 and RE 2) were printed with Ag|AgCl ink. The paper-based device had three functional areas. The CEs and REs were printed in the blue area. The microfluidic channels (MC 1 and MC 2) and filter holes (FH 1 and FH 2) were on the gray area. The working electrodes were printed in the red area. The working electrodes had a diameter of 3 mm, the microfluidic channels had a width of 1.5 mm, and the filter holes had a diameter of 4.5 mm. The folding of the device is shown in Fig. 7b. The sample

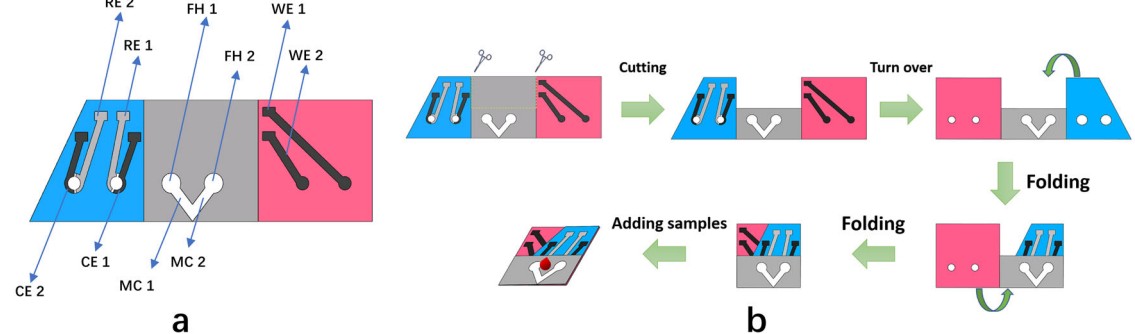

**Fig. 7 Structure and fabrication of the origami paper-based device. a** Display for each functional area, including working electrodes (WE 1 and WE 2), counter electrodes (CE 1 and CE 2), reference electrodes (RE 1 and RE 2), microfluidic channels (MC 1 and MC 2), and filter holes (FH 1 and FH 2). **b** The folding process of the device. Cut along the dotted line and fold the paper-based device as shown in the picture to get an available device.

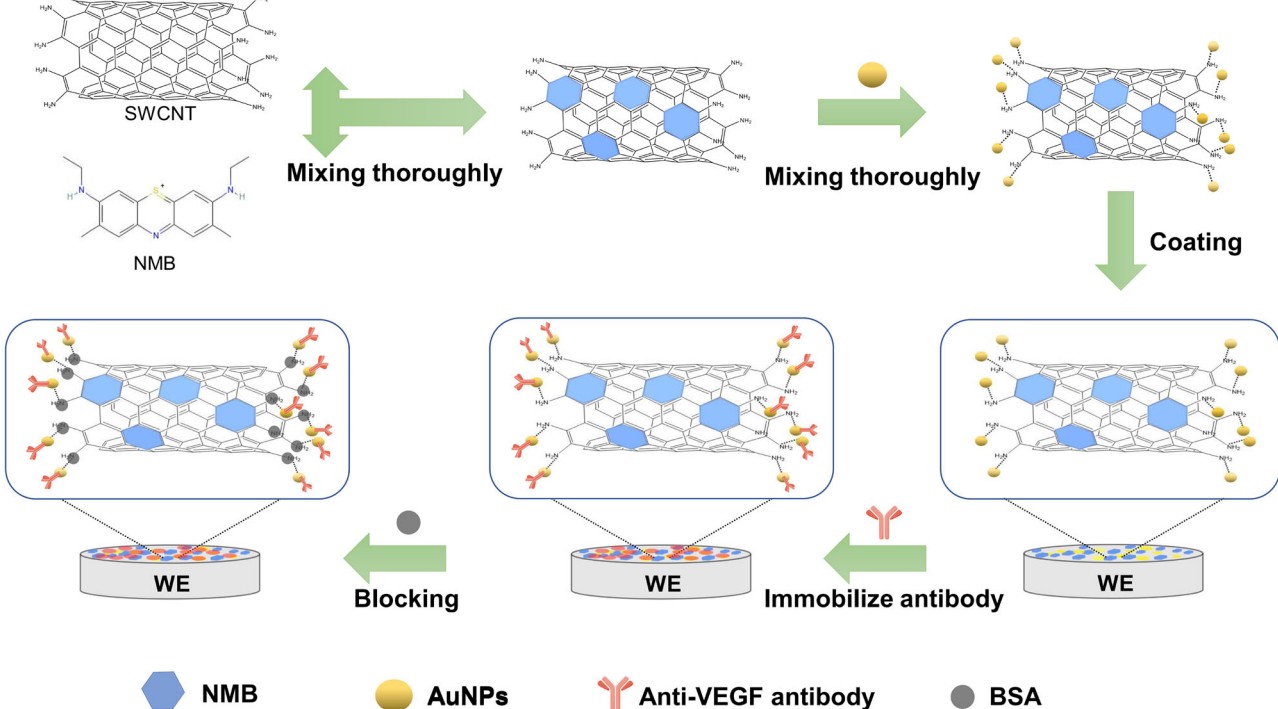

**Fig. 8 Nanocomposite modification process of the VEGF-C immunosensor.** Schematic diagram of nano material modification. First, the single wall carbon nanotubes (SWCNT) and new methylene blue (NMB) were mixed. After that, gold nanoparticles (AuNPs) were modified on the NMB/SWCNT nanocomposites. Then, the NMB/SWCNT/AuNPs three-in-one nanocomplex was assembled on the surface of the working electrode. After the antibody was fixed, the other sites on the electrode surface were blocked with BSA. Finally, the nanocomposites were successfully modified on the electrode surface.

drop flowed along the microfluidic channels. Running through the filter holes, it came into contact with the three electrodes. Then the sample could be detected electrochemically. Compared to traditional paper-based biosensors, the origami device had two main advantages. First, paste-free technology made the device easy to use. Before measuring, the user only needed to fold the paper along the black line. Once the device was folded, the electrode interface was naturally exposed, which reduced the difficulty of use. Second, all electrodes were printed on the front side, which decreased the amount of printing masks, which in turn reduced production costs.

**Synthesis of functional biologically modified materials**. A diagram of the preparation of the paper-based VEGF-C biosensor is shown in Fig. 8. To recap briefly, we synthesized NMB/NH$_2$-SWCNT/AuNP nanocomposites coated on the surface of working electrodes to achieve VEGF-C protein detection. The process of synthesizing NMB/NH$_2$-SWCNT/AuNP nanocomposites was as follows:

We first prepared NMB/NH$_2$-SWCNT composites. That is, 2 mg mL$^{-1}$ NMB was prepared, and simultaneously 2 mg mL$^{-1}$ NH$_2$-SWCNT was prepared. We mixed the two materials at a ratio of 1:1. To ensure that the composites were well mixed, we stirred the mixture vigorously for 24 h at 25 °C. Next 100 μL AuNP solution was mixed with 20 μL NMB/NH$_2$-SWCNT nanocomposite obtained previously. The AuNP solution mentioned above were synthesized in our laboratory. The protocol of the synthesis was as follow: Firstly, 1% Chloroaurate solution and 1% trisodium citrate solution were prepared. And then, 0.5 ml 1% chloroauric acid solution was added into 49.5 ml ultra-pure water to reduce the concentration of chloroauric acid to 0.01%. After that, the solution was heated to boiling with a magnetic stirrer. Under the vigorous stirring of the magnet, 2.5 ml of trisodium citrate solution was added rapidly. Subsequently, the mixture was stirred vigorously for 20 min. Then stopped heating after the solution turned red. Continued stirring until the temperature of the solution fell back to room temperature. Through this method we obtained nano-gold particles with a diameter of about 20 nm.

After synthesizing the NMB/NH$_2$-SWCNT/AuNP nanocomposite, we modified the working electrode with 10 μL of the composite. Working electrodes were placed in an oven (50 °C) until completely dried. Subsequently, 10 μL anti-VEGF antibody was added to the working electrodes. Because of interactions between the AuNPs and the sulfhydryl group of antibodies, they made an excellent combination with the working electrodes. After drying at 4 °C, 10 μL 1% BSA protein solution, whose solvent was a PBS solution, was dropped on the surface of the working electrodes as a blocking solution. Finally, the device was stored at 4 °C.

**Immunoassay of the paper-based immunosensor**. Prior to detecting the standard sample, we needed to wash the working electrodes several times with PBS solution. Having done that, we dropped 10 μL of different concentrations of VEGF-C protein on the device. To ensure that the proteins and antibodies were fully integrated, we incubated the device at 25 °C for 20 min. Then we used an autolab instrument to evaluate the performance of the device by detecting CV and DPV properties. Because the shape of the VEGF-C protein concentration can influence the peak current of a DPV curve, we could calculate the concentrations of samples by examining their peak current. We determined CV curves at a scan rate of 100 mV/s between −0.6 and 0.1 V. At the same time, we measured DPV responses under conditions of 0.025 s modulation time, 5 mV step potential and 0.5 s interval time in the range of −0.6 to 0.1 V.

**Statistics and reproducibility**. The TEM and SEM images were repeated at least once at a different time or by a different researcher. The electrochemical properties (Fig. 2) and stability (Fig. 6) of the immunosensor were successfully reproduced. The error bars shown in the text (Fig. 3, Fig. 4 and Fig. 5) were derived from three independent experiments. The analytical results of the VEGF-C clinical serum samples were verified by clinical application specialists.

**Reporting summary**. Further information on research design is available in the Nature Research Reporting Summary linked to this article.

## Data availability
The authors declare that the data supporting the findings of this study are available within the paper and the supplementary information files. Source data for charts in the main figures is available in Supplementary Data 1.

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

## Acknowledgements

This work was sponsored by the National Natural Science Foundation of China (NSFC) (61960206012, 61527815, 61775216, 61673024, 61771452), the National Key Research and Development Program (2017YFA0205902), and the Key Research Programs (QYZDJ-SSW-SYS015) of Frontier Sciences, CAS.

## Author contributions

S.S. and Y.W. designed and implemented the experiment. J-T.L., J-P.L., and X-X.C. provided technical assistance. Y.X. and T.M. performed the laboratory work. Y.X., S.Y., Y-Y.M., and Y.Y. contributed the clinical serum sample from the hospital. S.S. and Y.W. wrote the paper, and all authors discussed the results and commented on the manuscript.

## Competing interests

The authors declare no competing interests.
