## [Peer Review File · Communications Biology]

Reviewers' comments:

Reviewer #1 (Remarks to the Author):

Comments and Suggestions for Authors

In this work, the authors reported the use of origami 2 paper-based electrochemical immunosensor for highly specific detection of 3 VEGF-C using SWCNT/New methylene blue/Gold nanoparticles as a sensor. The report shows good results obtained and good organized experiment and presentation. Therefore, this study looks good and may be accepted after addressing the following comments;

1. From reading, I cannot be motivated from using new methylene blue. Why have to be new methylene blue? How does it better than methylene blue or other materials?
2. The function of each material used in sensor must be clearly discussed to see the importance of using these materials
3. For optimization of experimental conditions including concentration of NMB, the amount of AuNPs and the concentration of the antibody, the authors reported the optimal values that provided the best signal in detection. Anyway, the discussion for support these selection is really importance for readers.
4. For investigation of selectivity, the authors mixed 1 ng mL⁻¹ VEGF-C protein with 10 ng mL⁻¹ of the interfering substance. After that, the authors determined the DPV response with the VEGF-C immunosensor. I have noticed that current variation from interfering substances was 7.88%, -4.55%, 168 2.48%, 3.04%, -3.18%, and 1.16%. Therefore, what is the criteria used for adjustment that there is no effect from interference? It needs to be discussed.
5. For real sample analysis, what is the maximum concentration of possibly found interference?
6. How can control the performance between sensor and sensor? And what is the reproducibility between sensor and sensor?
7. There is no comparison with previous work. The authors must discuss the advantages of their work over the previous.

Reviewer #2 (Remarks to the Author):

The authors describe in this manuscript the development of an origami paper based immunosensor for the electrochemical detection of the VEGF-C biomarker in blood serum.

The manuscript is well written and the analytical performances of the immunosensors seems suitable for the detection of this metabolite in serum samples.

However, the detection principle, so as the workflow and the scheme of the manuscript are very similar to previous papers of the group, suggested by the authors in the Introduction as reference n. 31, but especially to reference n. 38 (same authors), where the only change, in addition to the target analyte of course, seems to be the substitution of the mediator thionine with the methylene blue used in this paper, electrochemical mediator well known and largely adopted in literature.

The only novelty of this work seems related to the design of the origami paper-based support, where two 3-electrodes sensors are printed. The two sensors seems both working, being connected by the filter holes, but no mention of the reason for the use of two sensors, nor to their behaviour can be found in the manuscript. Other minor points are:

- the authors refer to the mediator as "new methylene blue", but in what is it new? Please, clarify this point;
- what the source and characteristics of the AuNPs?
- on pg. 10, line 302, "NMB/NH₂-SWCNT/AuNP nanocomposite obtained previously" should be without AuNP, i.e. "NMB/NH₂-SWCNT".

Reviewer #3 (Remarks to the Author):

The manuscript describes a paper-based electrochemical immunosensor which was prepared by the NMB/NH₂-SWCNT/AuNP composites to detect blood VEGF-C. My suggestion is that this paper can be reconsidered after major revision to the journal after careful revision.

Specific comments:

(1) The authors write that: "SWCNT, which had a tubular structure. NMB molecules could combine with SWCNT by π - π stacking interactions. Meanwhile, circular particles adhered evenly to the SWCNT. This suggests that AuNPs were uniformly distributed on single-walled carbon nanotubes without aggregation by the connection between Au and amino groups." But in Figure 1B, the SEM images of the nanocomposite are unclear can not verify this statement.

(2) The authors haven't given the justification for using the complicated NMB/NH₂-SWCNT/AuNP materials. Why the three-in-one materials were better than other graphene or SWNTs-based materials?

(3) Strictly speaking, electrochemical experiment is lacked to characterize the assembly process of VEGF-C sensor.

(4) The manuscript describes: "comparison the results of clinical serum sample detected by a commercial instrument and the VEGF-C immunosensor. The results demonstrate that the paper-based device had an excellent specificity of detection. There is no data to support this conclusion just according the Table 1. Moreover, other VEGF-C sensors should be given for comparison.

(5) Whatman No. 1 Chromatography Paper was used as the substrate of the paper-based biosensor. Is there any special characteristic was requirement for paper selected?

(6) When it comes to paper-based analytical device, the following reference can be cited:

Li et al., *Sensors and Actuators B*, 2016, 231: 230–238.

Wang et al., *Electrochimica Acta*, 2016, 204: 128–135.

Li et al., *J. Mater. Chem. B*, 2014, 2: 6669–6674.

Reply to Reviewers' comments:

1、 Reply to Reviewer #1:

Comments of Reviewer #1:

In this work, the authors reported the use of origami paper-based electrochemical immunosensor for highly specific detection of VEGF-C using SWCNT/New methylene blue/Gold nanoparticles as a sensor. The report shows good results obtained and good organized experiment and presentation. Therefore, this study looks good and may be accepted after addressing the following comments;

Response: We sincerely thank the reviewer for affirmation to the manuscript, including experiment, results, and the presentation, which has great guiding significance for our work.

Q1: From reading, I cannot be motivated from using new methylene blue. Why have to be new methylene blue? How does it better than methylene blue or other materials?

Response: We thank the referee to raise this important issue. New methylene blue (NMB) is an electroactive material with excellent redox characteristics, which could generate electrochemical currents through the redox reactions on the surface of the electrodes. Moreover, once the corresponding antigen and antibody formed into immunocomplex, they would block the surface adsorption process of NMB molecules for the subsequent redox reactions, resulting a lower current.

We have to admit that some other electroactive materials, such as Prussian blue¹ and Thionine² could also be used in such circumstances for detection. However, our ultimate goal is to do multiplexed detection based on a single paper-based device. As a result, various electroactive materials with different oxidation peaks are needed to distinguish between the detection targets. These are the reasons why we use NMB in this work.

Q2: The function of each material used in sensor must be clearly discussed to see the importance of using these materials.

Response: We thank the referee to raise this important issue. Based on your valuable comments, we have discussed the functions of the three materials, and we have added the discussions into the 'Discussion' section. These sentences were highlighted in red (Page 9 to 10). In this work, three kinds of materials, including new methylene blue (NMB), single-walled carbon nanotubes (SWCNT), and gold nanoparticles (AuNPs) are used to modify the working electrode. As we mentioned in the manuscript, NMB can introduce electrochemical currents for the immunosensor due to its redox characteristics. SWCNT is used because of the following two reasons: i) it could improve the detection sensitivity due to its large specific surface area and good electrical conductivity; ii) it could help immobilize NMB through π - π stacking interactions. In addition, the AuNPs are introduced not only for its fast electron transportation but also for the interactions between amino groups of VEGF-C antibodies and gold. In the end, these three materials are synthesized together to form a functionalized nanomaterial.

Q3: For optimization of experimental conditions including concentration of NMB, the amount of AuNPs and the concentration of the antibody, the authors reported the optimal values that provided the best signal in detection. Anyway, the discussion for support these selection is really importance for readers.

Response: Thanks to the reviewer's suggestions. The discussions of optimization of experimental conditions have been added in the "Discussion" section. And these sentences are highlighted in red (Page 10).

Q4: For investigation of selectivity, the authors mixed 1 ng mL⁻¹ VEGF-C protein with 10 ng mL⁻¹ of the interfering substance. After that, the authors determined the DPV response with the VEGF-C immunosensor. I have noticed that current variation from interfering substances was 7.88%, -4.55%, 2.48%, 3.04%, -3.18%, and 1.16%. Therefore, what is the criteria used for adjustment that there is no effect from interference? It needs to be discussed.

Response: We thank the referee to raise this important issue. This is a very construction question, which is of great help to our follow-up work. Our way of expression may make you confused, so we explain this issue as follow. We have found some research to explore the criteria for selectivity. Yang's group from the University of Glasgow fabricated an aptasensor for the detection of PSA³. The concentration of interference and the target substance were both 1 ng mL⁻¹. The influence of the interfering substances was no more than 7.24% on the aptasensor in their work. In the meanwhile, Yang's group from the Cranfield University proposed an aptasensor for the estimate of the waste water. They also did a lot of work on selectivity. The influences of the interference were all below 12.9% in their work⁴. In our research, when the concentration of interference was ten times that of the target protein, the influence of the interference mentioned in our work does not exceed 7.88% on the sensor. Therefore, according to

figure 4, our paper-based device also had a good performance on selectivity.

Q5: For real sample analysis, what is the maximum concentration of possibly found interference?

Response: We thank the referee to raise this important issue. In this work, we selected six kinds of interference substances, including PD-L1, CEA, NSE, EGFR, UA and AA to test the selectivity of the immunosensor. According to the references, the maximum concentration of the above-mentioned substances possibly found during real sample analysis are shown as follows:

The concentration of PD-L1 is less than 107.2 pg mL^{-1} in the normal serum sample⁵. The concentration of CEA is no more than 10 ng mL^{-1} in the serum of healthy people⁶. The normal value of the concentration of NSE is in the range of 2.2 ng mL^{-1} to 10.2 ng mL^{-1} ^{7,8}. The range of EGFR biomarker in healthy individuals is in the range of about 1.0 to 25 ng mL^{-1} ⁹. The mean serum concentration of UA in the normal group is $43.9 \pm 12.4 \text{ } \mu\text{g mL}^{-1}$ ¹⁰. The concentration of serum ascorbic acid is $9.48 \pm 6.45 \text{ } \mu\text{g mL}^{-1}$ in healthy subjects¹¹.

Q6: How can control the performance between sensor and sensor? And what is the reproducibility between sensor and sensor?

Response: We thank the referee to raise this highly important issue. To improve the reproducibility of the immunosensor, we firstly optimized the fabrication process of the paper-based device to make sure that the unmodified electrodes exhibited the same performances. Moreover, the electrochemical properties are mainly based on the nanomaterials. As a result, we strictly controlled the synthetic process, including the mixing time, mixing ratio, and the concentrations of each materials.

The reproducibility between sensors refers to the difference in peak currents between several sensors for the detection of a specific concentration of VEGF-C. In this experiment, three different VEGF-C immunosensors were used for the detection of VEGF-C protein whose concentration was 500 pg mL^{-1} . The result showed that coefficient variation of the device is 0.67%.

Q7: There is no comparison with previous work. The authors must discuss the advantages of their work over the previous.

Response: We thank the referee to raise this important issue. The comparison result with previous work is really important for readers, so we have added the results into the 'Results' section, and highlighted

the sentences in red (Page 9). We have found some reported works on detection of VEGF. The comparisons on properties, including linear range, limit of detection and relative deviation are shown in the Table below. The results demonstrate that the origami paper-based device proposed in this work enables a relative wide linear range, with similar limit of detection.

Table 2 Comparison results of the proposed VEGF-C immunosensor and other sensors for detection of VEGF

No.	Substance	Linear Range (pg mL ⁻¹)	LOD (pg mL ⁻¹)	Relative deviation (%)	Reference
1	VEGF	10 -70	10	5.0	(Mustafa, 2011) ¹²
2	VEGF	0 - 400	100	10	(Ravalli, 2014) ¹³
3	VEGF	5 -125	5	1.47	(Mustafa, 2012) ¹⁴
4	VEGF ₁₆₅	10 - 800	12		(Moghadam, 2019) ¹⁵
5	VEGF-C	0.4 - 100U	0.4U	8.0	(Duff, 2003) ¹⁶
6	VEGF-C	10 - 100000	10	0.67	This work

2、 Reply to Reviewer #2:

Comments of Reviewer #2:

The authors describe in this manuscript the development of an origami paper based immunosensor for the electrochemical detection of the VEGF-C biomarker in blood serum.

The manuscript is well written and the analytical performances of the immunosensors seems suitable for the detection of this metabolite in serum samples.

However, the detection principle, so as the workflow and the scheme of the manuscript are very similar to previous papers of the group, suggested by the authors in the Introduction as reference No.31, but especially to reference No.38 (same authors), where the only change, in addition to the target analyte of course, seems to be the substitution of the mediator thionine with the methylene blue used in this paper, electrochemical mediator well known and largely adopted in literature. The only novelty of this work seems related to the design of the origami paper-based support, where two 3-electrodes sensors are printed. The two sensors seems both working, being connected by the filter holes, but no mention of the reason for the use of two sensors, nor to their behaviour can be found in the manuscript.

Response: We sincerely thank the reviewer for raising the key issue. This is not only a good comment, but also an important suggestion, which has vital guiding significance for us to summarize our works. According to the comments, we deeply analyzed the innovation of this work and further clearly stated the innovation points.

VEGF-C is an important biomarker in predicting advanced disease and individuals at risk for cancer. However, detection of VEGF-C is rarely reported up to now. In this work, 1) we have designed and fabricated a novel kind of dual-channel paper-based microfluidic device with two sets of electrodes. We can detect two targets simultaneously by using this device. Adopting an origami structure, this paper-based device further simplified the fabrication and operation steps, which had a high clinical significance. However, we have to admit that we only detected a single target for quick proof of concept in this work. In our next work, we will use this paper-based device to conduct dual-parameter testing for clinical needs, including but not limited to c-reactive protein (CRP), prealbumin (PAB), and other tumor markers, which can make clinical diagnosis more accurate; 2) we have made a new attempt in the selection of nanomaterials. Before detection, we optimized the synthesis conditions of the bio-sensitive membrane, including VEGF-C antibody concentration, electroactive substance concentration, and the mixing ratio between nanomaterials. After optimization of conditions, the immunosensor has obtained relatively good performance with a relatively high sensitivity, which made the device have the ability for the detection of VEGF-C; 3) the proposed low-cost paper-based platform, combining with the electrochemical detection method, could realize the goal of point-of-care and sensitive detection of VEGF-C.

Other minor points are:

Q1: The authors refer to the mediator as “new methylene blue”, but in what is it new? Please, clarify this point;

Response: We thank the referee to raise this important issue. New methylene blue is an organic compound of the thiazine class of heterocycles. It is closely related to methylene blue, which also belongs to a kind of electroactive material. However, its chemical formula is $C_{36}H_{44}Cl_4N_6S_2Zn$, which is different from methylene blue ($C_{16}H_{18}ClN_3S$).

Q2: What the source and characteristics of the AuNPs?

Response: We thank the referee to raise this important issue. The synthesis process of gold nanoparticles is indeed very attractive to readers, so we have added this part of the content to the 'Methods' section, and these sentences have been marked in red (Page 12). The AuNPs mentioned in the manuscript were synthesized in our laboratory. The protocol of the synthesis was as follow: Firstly, 1% chloroaurate solution and 1% trisodium citrate solution were prepared. And then, 0.5 ml 1% chloroauric acid solution was added into 49.5 ml ultra-pure water to reduce the concentration of chloroauric acid to 0.01%. After that, the solution was heated to boiling with a magnetic stirrer. Under the vigorous stirring of the magnet, 2.5 ml of trisodium citrate solution was added rapidly. Subsequently, the mixture was stirred vigorously for 20 minutes. Then stopped heating after the solution turned red. Continued stirring until the temperature of the solution fell back to room temperature. Through this method we obtained nano-gold particles with a diameter of about 20 nm.

Q3: On pg. 10, line 302, "NMB/NH₂-SWCNT/AuNP nanocomposite obtained previously" should be without AuNP, i.e. "NMB/NH₂-SWCNT".

Response: We thank the referee to raise this important issue. Now we have corrected the mistake in the revised manuscript and marked them in red (Page 12).

3、 Reply to Reviewer #3:

Comments of Reviewer #3:

The manuscript describes a paper-based electrochemical immunosensor which was prepared by the NMB/NH₂-SWCNT/AuNP composites to detect blood VEGF-C. My suggestion is that this paper can be reconsidered after major revision to the journal after careful revision.

Response: We sincerely thank the reviewer for affirmation to the manuscript. In response to the reviewer's suggestions, we have added experiments and answered the comments as good as we can. Your suggestions have played an important guiding significance for our future work.

Specific comments:

Q1: The authors write that: “SWCNT, which had a tubular structure. NMB molecules could combine with SWCNT by π - π stacking interactions. Meanwhile, circular particles adhered evenly to the SWCNT. This suggests that AuNPs were uniformly distributed on single-walled carbon nanotubes without aggregation by the connection between Au and amino groups.” But in Figure 1B, the SEM images of the nanocomposite are unclear can not verify this statement.

Response: We thank the referee to raise this important issue. To address this problem, we have added another set of SEM images which can verify the statement. Then we changed the images of figure 1B (Page 3). We can see that after modification with the synthesized nanocomposites, many nano-porous structures, which are distributed uniformly, are introduced onto the electrode surface. These high effective surface areas will be beneficial for allowing a higher density load of electrochemically active molecules and AuNPs.

Q2: The authors haven't given the justification for using the complicated NMB/NH₂-SWCNT/AuNP materials. Why the three-in-one materials were better than other graphene or SWNTs-based materials?

Response: Thank you for your kindly suggestions and professional comment very much. Each component in NMB / NH₂-SWCNT / AuNP has its own function, and all three components are essential. As we mentioned in the manuscript, NMB can introduce electrochemical currents for the immunosensor due to its redox characteristics. SWCNT is used because of the following two reasons: i) it could improve the detection sensitivity due to its large specific surface area and good electrical conductivity; ii) it could help immobilize NMB through π - π stacking interactions. In addition, the AuNPs are introduced not only for its fast electron transportation but also for the interactions between amino groups of VEGF-C

antibodies and gold. In the end, these three materials are synthesized together to form a functionalized nanomaterial. It can be seen that only graphene or single-walled carbon nanotube sensors cannot provide redox potential and assemble antibodies on the immunosensor. Therefore, in order to make an electrochemical immunosensor, a three-in-one material is necessary.

Q3: Strictly speaking, electrochemical experiment is lacked to characterize the assembly process of VEGF-C sensor.

Response: We sincerely thank you for this insightful comment. The characterization of the assembly process of VEGF-C sensor was shown as follow. This part of words were added in the 'Results' section and highlighted in red (Page 4 to 5).

Fig. 2 Electrochemical characterization of the nanocomposites assembly process of VEGF-C sensor. (C) CV responses of the device: (c1) CV responses of $\text{NH}_2\text{-SWCNT/NMB/AuNP}$ nanocomposites modified working electrode; (c2) The working electrode immobilized with $\text{NH}_2\text{-SWCNT/NMB}$ composites; (c3) The working electrode coated with NMB only. (D) DPV response of the device: (d1) The working electrode modified with $\text{NH}_2\text{-SWCNT/NMB/AuNP}$ nanocomposites; (d2) The working electrode immobilized with $\text{NH}_2\text{-SWCNT/NMB}$ composites; (d3) The working electrode coated with NMB only.

The CV and DPV responses were measured with an autolab instrument in 0.1 M PBS solution ($\text{pH} = 7.4$). The CV responses are shown in Fig. 2(C). From this graph, we can see that the working electrode had an electrochemical signal due to the fact of NMB (curve c3). By contrast, after adding the $\text{NH}_2\text{-SWCNT}$ nanomaterial, the electrode showed a better electrochemical performance, which could suggest that $\text{NH}_2\text{-SWCNT}$ could reduce the resistance of the sensor (curve c2). Similarly, when AuNP was assembled on the nanocomposites, the peak current also increased gradually (curve c1).

The obtained DPV responses are shown in Fig. 2(D). The responses were measured under the same conditions as the CV responses. As shown in Fig. 2(D), when the NMB was modified on the working electrode, the sensor had a redox signal (curve d3). When NH₂-SWCNT was connected to NMB, the peak current rose to 22.3 μ A (curve d2). Then, when AuNP was assembled on the nanocomposites, the working electrode (curve d1), whose peak current was 25.91 μ A, performed better than the sensor modified by NH₂-SWCNT/NMB composites, which confirmed that the gold nanoparticles could improve the detection sensitivity of the sensor.

Q4: The manuscript describes: “comparison the results of clinical serum sample detected by a commercial instrument and the VEGF-C immunosensor. The results demonstrate that the paper-based device had an excellent specificity of detection. There is no data to support this conclusion just according the Table 1. Moreover, other VEGF-C sensors should be given for comparison.

Response: We thank the referee to raise this important issue. We are so sorry that our interpretation of Table 1 makes you confused. Table 1 compared the detection results of the sensors mentioned in this manuscript with the ELISA test results. Since ELISA is a commercial testing method, the results of the ELISA test have certain authority. Therefore, we believed that the ELISA test results of clinical serum samples were accurate values (reference concentration). At the same time, we called the testing results of our immunosensors the proposed concentration. Then, we compared the reference concentrations with the proposed concentration. After calculating, we obtained a relative deviation. Relative deviation showed the detection error of the immunosensor. According to Table 1, relative deviation was small which suggested that the immunosensor had good detection specificity.

We thank the referee to raise the second issue. The comparison result with previous work is really important for readers, so we have added the results into the ‘Results’ section, and highlighted the sentences in red (Page 9). We have found some reported works on detection of VEGF. The comparisons on properties, including linear range, limit of detection and relative deviation are shown in the Table below. The results demonstrate that the origami paper-based device proposed in this work enables a relative wide linear range, with similar limit of detection.

Table 2 Comparison results of the proposed VEGF-C immunosensor and other sensors for detection of VEGF

No.	Substance	Linear Range (pg mL ⁻¹)	LOD (pg mL ⁻¹)	Relative deviation (%)	Reference
1	VEGF	10 -70	10	5.0	(Mustafa, 2011) ¹²

2	VEGF	0 - 400	100	10	(Ravalli, 2014) ¹³
3	VEGF	5 -125	5	1.47	(Mustafa, 2012) ¹⁴
4	VEGF ₁₆₅	10 - 800	12		(Moghadam, 2019) ¹⁵
5	VEGF-C	0.4 - 100U	0.4U	8.0	(Duff, 2003) ¹⁶
6	VEGF-C	10 - 100000	10	0.67	This work

Q5: Whatman No. 1 Chromatography Paper was used as the substrate of the paper-based biosensor. Is there any special characteristic was requirement for paper selected?

Response: We thank the referee to raise this important issue. For the selection of paper, we consider the following three factors. First, the thickness of the paper should be appropriate. If the paper is too thick, it will cause poor liquid flow and hinder the penetration of the sample between the papers. If the paper is too thin, it will lead to a bad effect of screen printing; Second, the hardness of the paper should be appropriate. If the hardness of the paper is too large, it will increase the difficulty of folding the paper. Third, the pore size of the filter paper should be appropriate. An excessively large pore size will affect the filtering effect of the sensor filter layer. However, the too small pore size will decrease the solution transfer rate between the folded layers. Therefore, based on the above three factors, we chose Whatman No.1 filter paper as the substrate, the pore size of the filter paper is 11 μm, which can filter out part of the blood cells, and have the ability to complete the function of microfluidic channel and folding in actual operation.

Q6: When it comes to paper-based analytical device, the following reference can be cited:

Li et al., Sensors and Actuators B, 2016, 231: 230–238.

Wang et al., Electrochimica Acta, 2016, 204: 128–135.

Li et al., J. Mater. Chem. B, 2014, 2: 6669–6674.

Response: Thanks to the reviewer's suggestions, the work of these three articles is excellent, which has a good guiding role for the article. These three articles have been added to the references (Page 2 and Page 15).

Reference

1. Li NS, *et al.* Mobile healthcare system based on the combination of a lateral flow pad and smartphone for rapid detection of uric acid in whole blood. *Biosensors & Bioelectronics* **164**, (2020).
2. Wang Y, *et al.* Electrochemical integrated paper-based immunosensor modified with multi-walled carbon nanotubes nanocomposites for point-of-care testing, of 17 beta-estradiol. *Biosensors & Bioelectronics* **107**, 47-53 (2018).
3. Wei B, Mao K, Liu N, Zhang M, Yang ZJB, *Bioelectronics*. Graphene nanocomposites modified electrochemical aptamer sensor for rapid and highly sensitive detection of prostate specific antigen. **121**, 41-46 (2018).
4. Mao K, Ma J, Li X, Yang ZJTeotte. Rapid duplexed detection of illicit drugs in wastewater using gold nanoparticle conjugated aptamer sensors. **688**, 771 (2019).
5. A TAH, A MF, B ZB, A GCJCRiOH. The clinical significance of soluble PD-1 and PD-L1 in lung cancer. **143**, 148-152 (2019).
6. Park JW, Chang HJ, Kim BC, Yeo HY, Kim DYJCD. Clinical validity of tissue carcinoembryonic antigen expression as ancillary to serum carcinoembryonic antigen concentration in patients curatively resected for colorectal cancer. **15**, e503-e511 (2013).
7. Casmiro M, Scarpa E, Cortelli P, Vignatelli LJC. Cerebrospinal fluid and serum neuron-specific enolase in acute benign headache. **28**, 506-509 (2010).
8. Schmitt B, *et al.* Serum and CSF levels of neuron-specific enolase (NSE) in cardiac surgery with cardiopulmonary bypass: a marker of brain injury? **20**, 536 (1998).
9. Kumar RR, Meenakshi A, Sivakumar NJHa. Enzyme immunoassay of human epidermal growth factor receptor (hEGFR). **10**, 143-147 (2001).

10. Karaosmanoglu N, Karaaslan E, Cetinkaya PO. Evaluation of serum uric acid levels in patients with rosacea. *Archives of Dermatological Research* **312**, 447-451 (2020).
11. Bghmer T. Concentrations of the watersoluble vitamins thiamin, ascorbic acid, and folic acid in serum and cerebrospinal fluid of healthy individuals¹³.
12. Biosensors MKSJ, Bioelectronics. A new impedimetric biosensor utilizing VEGF receptor-1 (Flt-1): Early diagnosis of vascular endothelial growth factor in breast cancer. **26**, 4032-4039 (2011).
13. Ravalli A, Marrazza G, Rivas L, Escosuramuniz ADL, Merkoci AJLNiEE. Electrochemical Antibody-Aptamer Assay for VEGF Cancer Biomarker Detection. **268**, 175-178 (2014).
14. a MKS, B ZOUJAB. An impedimetric vascular endothelial growth factor biosensor-based PAMAM/cysteamine-modified gold electrode for monitoring of tumor growth. **423**, 277-285 (2012).
15. Moghadam FM, Rahaie MJB, Bioelectronics. A Signal-on Nanobiosensor for VEGF165 detection based on Supraparticle Copper Nanoclusters Formed on Bivalent Aptamer. (2019).
16. Duff SE, Li C, Renehan A, O'Dwyer ST, Kumar SJIJoO. Immunodetection and molecular forms of plasma vascular endothelial growth factor-C. **22**, 339 (2003).

REVIEWERS' COMMENTS:

Reviewer #1 (Remarks to the Author):

Comments and Suggestions for Authors: COMMSBIO-20-0117A

In this work, the authors reported on the development of SWCNT/New methylene blue/Gold nanoparticles modified origami paper-based electrochemical immunosensor for highly specific detection of VEGF-C and applied the proposed method to detect VEGF-C in clinical serum samples. The experiment and the study are look good and may be accepted after addressing the following comments;

1. The authors must clearly explain and discuss about the function of NMB on their detection method. Even though, they used to propose the use of NMB for their previous work.
2. For electrochemical properties of the VEGF-C immunosensor section. I am not clear that how can use bare working electrode as a blank control? Typically, the blank control must be NMB/NH₂-SWCNT/AuNP nanocomposite. Bare electrode is only used to compare the performance of working electrode in the presence of NMB and AuNPs.
3. For selectivity, the authors fixed the concentration ratio between VEGF-C and interference at 1:1. Why? And what is happened at low and higher this ratio?
4. What kind of standard curve used for the determination of VEGF-C in serum samples?
5. Are there any sample matrix for this proposed method?

Reviewer #3 (Remarks to the Author):

The authors answered all questions well, and I think the current status of this article can be accepted by the journal.

REVIEWERS' COMMENTS:

Reviewer #1 (Remarks to the Author):

Comments and Suggestions for Authors: COMMSBIO-20-0117A

In this work, the authors reported on the development of SWCNT/New methylene blue/Gold nanoparticles modified origami paper-based electrochemical immunosensor for highly specific detection of VEGF-C and applied the proposed method to detect VEGF-C in clinical serum samples. The experiment and the study are look good and may be accepted after addressing the following comments;

***Response:** We sincerely thank the reviewer for affirmation to the manuscript, including experiment, results, and the presentation, which has great guiding significance for our work.*

Q1. The authors must clearly explain and discuss about the function of NMB on their detection

method. Even though, they used to propose the use of NMB for their previous work.

***Response:** We thank the reviewer for raising the important issue. New methylene blue (NMB) is an electroactive material with excellent redox characteristics, which could generate electrochemical currents through the redox reactions on the surface of the electrodes. As we can see in figure 2(C), contrast with the bare electrode (curve a5), the working electrode had an electrochemical signal due to the fact of NMB (curve c3) which could verify the electrochemical function of NMB. In the meanwhile, when corresponding antigen and antibody formed into immunocomplex, they would block the surface adsorption process of NMB molecules for the subsequent redox reactions, resulting a lower current. Figure 5(A) could verify the feasibility of the detection principle well.*

Q2. For electrochemical properties of the VEGF-C immunosensor section. I am not clear that how can use bare working electrode as a blank control? Typically, the blank control must be NMB/NH₂-SWCNT/AuNP nanocomposite. Bare electrode is only used to compare the performance of working electrode in the presence of NMB and AuNPs.

***Response:** We thank the referee to raise this important issue. I am sorry that our description “the bare working electrode as a blank control” in the legend of Figure 2 makes you feel confused. As you say, the responses of bare working electrodes in Figure 2 (curve a5 and b5) are only used as a comparison to make sure that the working electrodes have been successfully modified by the nanocomposites. There is some ambiguity about the description of the “blank control”. As a result, we have deleted it in the revised manuscript. As we can see, in the following process of quantitative determination, the response of NMB/NH₂-SWCNT/AuNP nanocomposite is used as the real blank control, which is consistent with the linear fitting curve we get.*

Q3. For selectivity, the authors fixed the concentration ratio between VEGF-C and interference at 1:1. Why? And what is happened at low and higher this ratio?

***Response:** We thank the referee to raise this important issue. We are sorry that the description of selectivity made you confused. The concentration of VEGF-C protein was 1 ng mL⁻¹ and the concentration of the interference was 10 ng mL⁻¹. The ratio of the concentration between VEGF-C and interfering substances was 1:10. After that, we mixed the VEGF-C and the interference in a volume ratio of 1:1. We usually keep the volume ratio, and change the concentration of interference to estimate the selectivity of the device. If the immunosensor can maintain a high degree of specificity under the*

influence of a high concentration of interfering substances, we consider the sensor had a good performance in selectivity. In our research, when the concentration of the interference was 10 times than VEGF-C, the change of the currents was no more than 7.88 %. If we increase the concentration ratio between the interference and VEGF-C, the change of the peak currents will increase. On the contrary, when the ratio decreased, the detection result will be closer to the peak current without interference.

Q4. What kind of standard curve used for the determination of VEGF-C in serum samples?

***Response:** We thank the referee to raise this important issue. Before the determination of VEGF-C in serum samples, we selected several serum samples to calibrate the sensor. After calibration, we used the standard curve to test serum samples, and then compare it with the results of ELISA to obtain the relative deviation.*

Q5. Are there any sample matrix for this proposed method?

***Response:** We thank the referee to raise this important issue. The sample matrix for the detection of standard solution was PBS solution which was the common solvent in solution preparation. In the meanwhile, the serum sample was obtained from hospital, we did not use any solvents to process the serum samples.*

Reviewer #3 (Remarks to the Author):

The authors answered all questions well, and I think the current status of this article can be accepted by the journal.

***Response:** We sincerely thank the reviewer for affirmation to the manuscript.*